# Molecular switching in transcription through splicing and proline-isomerization regulates stress responses in plants

Frederik Friis Theisen [1,2], Andreas Prestel [2], Steffie Elkjær [1], Yannick H. A. Leurs [1], Nicholas Morffy[3], Lucia C. Strader [3], Charlotte O'Shea[1], Kaare Teilum [2], Birthe B. Kragelund [1,2] ✉ & Karen Skriver [1] ✉

The *Arabidopsis thaliana* DREB2A transcription factor interacts with the negative regulator RCD1 and the ACID domain of subunit 25 of the transcriptional co-regulator mediator (Med25) to integrate stress signals for gene expression, with elusive molecular interplay. Using biophysical and structural analyses together with high-throughput screening, we reveal a bivalent binding switch in DREB2A containing an ACID-binding motif (ABS) and the known RCD1-binding motif (RIM). The RIM is lacking in a stress-induced DREB2A splice variant with retained transcriptional activity. ABS and RIM bind to separate sites on Med25-ACID, and NMR analyses show a structurally heterogeneous complex deriving from a DREB2A-ABS proline residue populating *cis*- and *trans*-isomers with remote impact on the RIM. The *cis*-isomer stabilizes an α-helix, while the *trans*-isomer may introduce energetic frustration facilitating rapid exchange between activators and repressors. Thus, DREB2A uses a post-transcriptionally and post-translationally modulated switch for transcriptional regulation.

Transcriptional pathways regulate diverse biological processes such as stress responses and development, and they culminate in regulation of gene expression by transcription factors (TFs)[1]. For this, interactions between activation domains (ADs) of TFs and co-regulators are essential by ensuring the correct location of RNA polymerase II[2]. ADs are interchangeable and can bind unrelated co-regulators[3], and most ADs are intrinsically disordered with low sequence conservation[4,5]. This has spurred the idea that interactions between ADs and co-regulators are short-lived and non-specific with stochastic burial of hydrophobic residues[4,6,7]. The mediator multi-protein complex is an important co-regulator, responsible for facilitating interactions between the basal RNA polymerase II machinery and TFs. Studies of the interactions between mediator subunits and TFs contribute to unraveling the complex nature of TF:co-regulator interactions, and have revealed emerging themes of multivalency, coupled folding and

binding, and dynamic interfaces. Thus, dynamic and multivalent interactions were demonstrated for the interaction between ETV4 and the activator interaction domain (ACID) of mediator complex subunit 25 (Med25)[8], bivalency was shown for the interactions between Med25-ACID and VP16[9,10] and p53[11], and the interface between Med25-ACID and ETV5 was dynamic with the AD of ETV5 undergoing coupled folding and binding upon complex formation[12].

Plant Med25 is implicated in a range of biological processes spanning from plant development to hormone signaling and stress responses[13]. *Arabidopsis thaliana* Med25 physically interacts with abiotic-stress-associated TFs, including Drought Response Element Binding protein 2 A (DREB2A), through its ACID domain[14]. The *Arabidopsis dreb2a* and *med25* mutants both displayed increased sensitivity to salt stress. However, Med25 and DREB2A had opposite functions in response to drought with DREB2A and Med25 increasing and

[1]The REPIN and The Linderstrøm-Lang Centre for Protein Science, Department of Biology, University of Copenhagen, Copenhagen, Denmark. [2]Structural Biology and NMR Laboratory, Department of Biology, University of Copenhagen, Copenhagen, Denmark. [3]Department of Biology, Duke University, Durham, NC, USA. ✉e-mail: bbk@bio.ku.dk; kskriver@bio.ku.dk

decreasing drought resistance, respectively[14,15]. Together, Med25 and DREB2A also function in the repression of PhyB-mediated light signaling and thus integrate signals from different important regulatory pathways in plants[14].

The RCD1-SRO-TAF4 (RST) αα-hub domain[16–18] is responsible for negative regulatory interactions of *Arabidopsis* Radical-induced Cell Death1 (RCD1) with DREB2A[19,20]. Downregulation of RCD1 or loss of the RCD1-interacting region of DREB2A, as in a DREB2A splice variant (Fig. 1A), was required for DREB2A accumulation during heat stress and senescence and thus for DREB2A function under stress conditions[21]. DREB2A interacts with RCD1-RST using a short linear motif (SLiM), the RCD1-interaction motif (RIM)[21], which undergoes coupled folding and binding[20]. Recently, the SLiM context was shown to increase affinity by stabilizing the complex[22]. The same SLiM has previously been hypothesized to facilitate the interaction with *At*Med25-ACID[23], however, full-length DREB2A was necessary to achieve strong binding[24]. Together, this suggests that the disordered region surrounding the RIM is a functional hot spot (Fig. 1).

In this work, we explore features of the functional hot spot to obtain mechanistic insight into molecular switching in transcriptional regulation. We identified a SLiM, the ACID-binding motif (ABS), which in conjunction with the RIM forms a bivalent Med25-ACID-interacting region that is functionally regulated by alternative splicing. Complementary structural analyses revealed that DREB2A bound to a groove and a hydrophobic surface on the ACID domain, and chemical exchange saturation transfer (CEST) NMR showed that the Med25-ACID:DREB2A complex was structurally heterogenous due to a DREB2A proline residue occupying highly populated *cis* and *trans* isomers. This introduces proline isomerization as a player in mediator:TF complexes, facilitating the energetic frustration needed for regulator exchange, and adding an additional layer of regulatory functionality to a transcription factor switch unit.

## Results

### The Med25-ACID-binding region of DREB2A contains two SLiMs

To define a DREB2A core region needed for Med25-ACID binding, we initially produced a long[23] fragment spanning from Val151 to the C-terminus, constituting the entire C-terminal intrinsically disordered region (IDR) of DREB2A (Fig. 1A). The interaction was characterized using isothermal titration calorimetry (ITC) (Supplementary Fig. 1) and

for DREB2A$_{151-335}$ we obtained a $K_d$ of $540 \pm 40$ nM (Table 1, Fig. 1A), with thermodynamics indicating an enthalpically driven interaction ($\Delta H = -52.3 \pm 0.8$ kJ/mol) with an entropic penalty ($-T\Delta S = 16.5 \pm 1.0$ kJ/mol).

To narrow down the Med25-ACID-binding region, DREB2A was truncated from both termini. Based on considerations of the ID-profile and predicted secondary structure, three DREB2A fragments (DREB2A$_{195-276}$, DREB2A$_{234-335}$, and DREB2A$_{234-276}$) were produced, all showing the similar binding affinity and thermodynamic parameters (Table 1 and Fig. 1A). Further truncation of the C-terminus (DREB2A$_{234-272}$ and DREB2A$_{234-256}$) resulted in a significant reduction of binding affinity, increasing $K_d$ to $1.8 \pm 0.4$ μM and $6.0 \pm 0.2$ μM, respectively. The RIM (residues 259-269, Fig. 1) was thus involved, but not necessary for binding. Although the RIM contributed substantial interaction energy, it was not possible to measure the affinity of the RIM by itself (DREB2A$_{255-276}$) using ITC. We then produced the DREB2A$_{195-335,\Delta244-276}$ variant, which had an internal deletion of a region (Trp244-Gly276), absent in a DREB2A splice variant unable to bind RCD1[21] but still with transcriptional activity[25]. This variant bound Med25-ACID with reduced affinity ($K_d = 5.4 \pm 0.6$ μM). Comparison of DREB2A sequences from phylogenetically representative plant species revealed that DREB2A$_{234-276}$ contains two separate conserved regions, one corresponding to the RIM present in DREB2A from land plants[26], and a putative motif with the consensus sequence LxVxD[YFL][GS]W[PI] (Fig. 1B). This region was designated the ACID-binding SLiM (ABS).

Since aromatic and acidic residues are important for binding of ADs to co-regulators[27], we analyzed how their mutations affected Med25-ACID binding (Table 1, Supplementary Fig. 2). Changing Asp241, Tyr242, or Trp244 of the ABS to alanine resulted in either a four-fold decrease in affinity (D241 A and W244 A) or no detectable binding (Y242A). For Phe259, which is essential for DREB2A binding to RCD1-RST[20], substitution to alanine hardly affected Med25-ACID binding affinity. Substitution of Phe274 with alanine also had only a minor effect on binding, although Phe274 belongs to the Val273-Gly276 region which affected the affinity considerably when removed (Table 1). In conclusion, the ABS serves as the primary Med25-ACID binding motif, although the RIM region also contributes. Hence, the RIM was defined as the secondary Med25-ACID binding motif.

The AD of DREB2A was previously mapped to the C-terminal residues 254-335 in *Arabidopsis* protoplasts[25]. To examine whether the

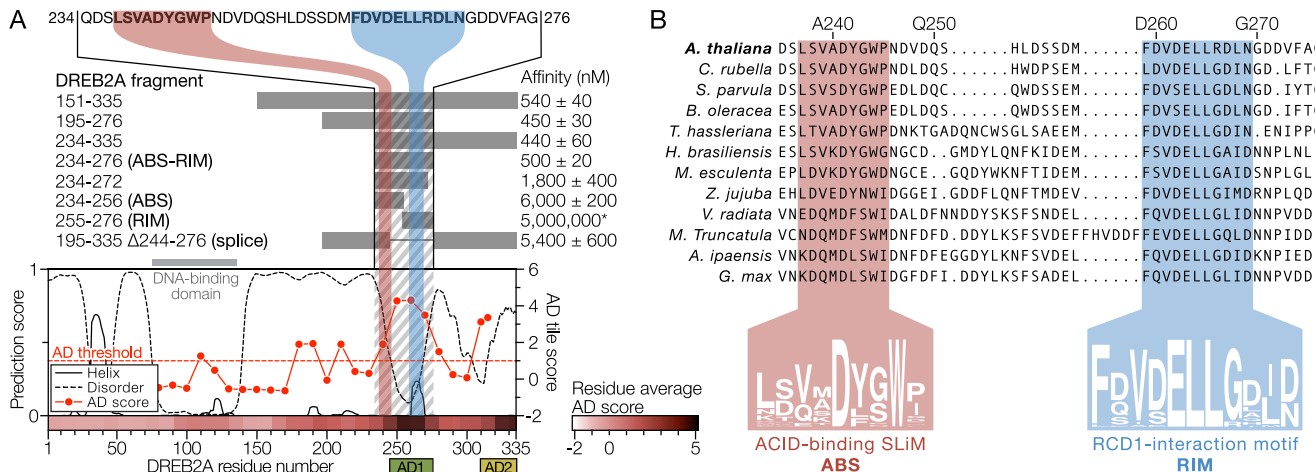

**Fig. 1 | Identification of the Med25-ACID binding region of DREB2A with a primary SLiM. A** Truncation of DREB2A and binding affinities determined using ITC. The graph shows disorder (DISOPRED) and helix propensity (Agadir) predictions of DREB2A. Red data points: AD score of 40-residue tiles as a function of the tile center position. Bottom heat map: residue specific AD scores calculated as the average score of all overlapping tiles. The higher AD score, the higher transcriptional activity. The hatched region is the minimal region needed to achieve maximum Med25 affinity. *Affinity determined using NMR. **B** Alignment of DREB2A orthologs from different plants species showing the position and consensus sequence of the ABS (red) and the RIM (blue). Source data are provided as a Source Data file.

**Table 1 | Thermodynamic parameters of DREB2A interaction with Med25-ACID at 25 °C**

| DREB2A | N-value | $K_d$ (nM) | $\Delta H$ (kJ/mol) | $-T\Delta S$ (kJ/mol) | $\Delta G$ (kJ/mol) |
|---|---|---|---|---|---|
| 151–335 | 1.10 ± 0.06 | 540 ± 40 | −52.3 ± 0.8 | 16.5 ± 1.0 | −35.8 ± 0.2 |
| 195–276 | 1.09 ± 0.02 | 450 ± 30 | −51.2 ± 0.6 | 15.0 ± 0.7 | −36.2 ± 0.2 |
| 234–335 | 1.0 ± 0.2 | 440 ± 60 | −58 ± 1 | 22 ± 2 | −36.3 ± 0.3 |
| 234–276 (ABS-RIM) | 0.98 ± 0.08 | 510 ± 20 | −50.7 ± 0.3 | 14.8 ± 0.4 | −35.9 ± 0.1 |
| 234–272 | 0.97 ± 0.05 | 1800 ± 400 | −43 ± 3 | 10 ± 4 | −32.8 ± 0.5 |
| 234–256 (ABS) | 0.86 ± 0.03 | 6000 ± 200 | −54.0 ± 0.5 | 24.2 ± 0.6 | −29.8 ± 0.1 |
| 255–276 (RIM)* | – | 5,000,000 | – | – | – |
| 195–335 Δ244–276 | 1.02 ± 0.09 | 5400 ± 600 | −59 ± 2 | 28 ± 2 | −30.1 ± 0.3 |
| **DREB2A 195–276** | | | | | |
| L237A | 1.03 ± 0.06 | 2400 ± 300 | −22.9 ± 0.7 | −9.2 ± 0.9 | −32.1 ± 0.3 |
| V239A | 0.95 ± 0.01 | 2000 ± 200 | −42 ± 1 | 10 ± 1 | −32.5 ± 0.3 |
| D241A | 1.07 ± 0.06 | 2000 ± 700 | −37 ± 7 | 5 ± 7 | −32.5 ± 0.9 |
| Y242A | – | NB | – | – | – |
| W244A | 1.0 ± 0.1 | 1900 ± 200 | −47 ± 1 | 14 ± 2 | −32.7 ± 0.3 |
| P245A** | 1.26 ± 0.01 | 780 ± 60 | −43.4 ± 0.8 | 8.8 ± 0.8 | −34.9 ± 0.2 |
| F259A | 1.2 ± 0.1 | 550 ± 50 | −58.8 ± 0.9 | 23 ± 1 | −35.8 ± 0.2 |
| E263P | 1.00 ± 0.06 | 580 ± 40 | −61.2 ± 0.8 | 25.5 ± 1.0 | −35.6 ± 0.2 |
| F274A | 0.86 ± 0.02 | 660 ± 90 | −62 ± 2 | 27 ± 2 | −35.3 ± 0.3 |
| **Med25-ACID** | | | | | |
| R568A | 0.90 ± 0.01 | 3,100 ± 700 | −46 ± 4 | 14 ± 5 | −31.5 ± 0.5 |

All DREB2A WT ITC data represent the results of two or more replicates. Shown values represent mean and standard deviations from replicates. *Affinity determined using NMR, **P245A variant was made in the DREB2A$_{234-276}$ fragment, NB: no detectable binding. The table is provided as a Source Data file.

RIM and ABS are both active in transcription, we analyzed the AD pattern using a high-throughput yeast-based assay based on synthetic TFs containing a DNA-binding domain, targeting a GFP reporter gene, an mCherry fluorescence tag and a variable putative AD (Supplementary Fig. 3). The screen was performed using 40-residue overlapping tiles spanning the entire DREB2A sequence with the AD score of each tile calculated from the GFP:mCherry fluorescence intensity ratio, thus reflecting the transcriptional activity[28]. This revealed that the C-terminal AD[25] contains two distinct ADs, AD1 (app. residues 250-270) and AD2 (app. residues 320-335) (Fig. 1A), defined as residues for which all tiles displayed an AD score above the threshold (Supplementary Fig. 3E). The RIM overlapped with AD1 whereas the ABS, although enhancing the effect of the RIM, did not by itself correlate with transcriptional activity.

### AtMed25-ACID forms a β-barrel with two accessory helices and dynamic regions

The structure of the human hMed25-ACID domain was previously determined using NMR spectroscopy[9,10]. However, sequence alignments of At- and hMed25-ACID showed low sequence identity[23] (Supplementary Fig. 4A) and the AlphaFold2 (AF2) prediction of the AtMed25-ACID structure (Supplementary Fig. 4B)[29,30] differed from the published hMed25-ACID structure. Both domains form a core β-barrel with accessory α-helices of various lengths, but hMed25-ACID featured an additional helix and a considerably longer loop 1 (L1). To validate the predicted structure of the AtMed25-ACID domain, the backbone NMR chemical shifts were assigned (Supplementary Fig. 5A). Secondary chemical shifts of $^{13}C^{\beta}$ nuclei showed positive signals suggesting a structure composed mostly of β-strands (Supplementary Fig. 5B). Secondary structure propensities (SSPs) were then quantified based on multiple nuclei using the motif identification from chemical shifts (MICS) server[31] (Fig. 2A) and mapped onto the AF2 structure model (Fig. 2B). This showed an excellent agreement between prediction and data which was further supported by small-angle X-ray scattering (SAXS) data validating the overall dimensions of the AtMed25-ACID domain ($\chi^2 = 1.008$, Supplementary Fig. 5C).

Using molecular dynamics (MD), the predicted structure was simulated for 1.7 μs to examine the dynamics of the Med25-ACID domain structure. Several regions exhibited larger fluctuations on the ns timescale (Fig. 2C and D). We therefore recorded backbone $\{^1H\}$-$^{15}N$ heteronuclear NOEs (hetNOEs), as well as $^{15}N$ transverse ($R_2$) and longitudinal ($R_1$) relaxation rates of AtMed25-ACID (Fig. 2C), which report on the ps to ns timescale dynamics of the backbone amides. These data provided some indication of increased dynamics of the loop regions, although chemical exchange contributions made definite conclusions difficult. The hetNOEs, which are not affected by chemical exchange, revealed a clearer pattern with increased dynamics (lower hetNOE) of the backbone amides in L1 and H1L3, and to a smaller degree in L2 and L4.

Thus, the analyses of secondary structure, overall shape, and dynamics of the AtMed25-ACID domain were in good agreement with the AF2 prediction, suggesting that the AF2 structure is a suitable model of the AtMed25-ACID domain.

### DREB2A binds a groove in AtMed25-ACID formed by H2 and L1

To identify the DREB2A binding sites in Med25-ACID, we performed $^{15}N$-HSQC titration experiments using $^{15}N$-labeled Med25-ACID. Mapping of $^{15}N,H^N$ chemical shift perturbations (CSPs), induced upon addition of ABS (DREB2A$_{234-256}$, Fig. 1A) (Fig. 3A), revealed CSPs localized around a groove defined by the dynamic L1 and the H2 α-helix (Figs. 2D and 3B-D). The largest CSPs were seen around L1 while the effects in H2 were smaller, indicating that L1 undergoes structural changes, while H2 remains relatively unaffected. To examine the bound state dynamics of L1 we recorded $\{^1H\}$-$^{15}N$ hetNOEs of Med25-ACID in complex with the ABS (Supplementary Fig. 6A). Comparison of the hetNOEs of the folded ACID domain in the free and ABS bound states revealed two regions with significant differences, one of which mapped to L1 and the other to the C-terminus of H2.

To validate the proposed binding groove, we introduced an alanine in L1, replacing Arg568, which experienced a large CSP upon DREB2A binding, and determined the binding affinity using ITC with the high-affinity ABS-RIM fragment (DREB2A$_{234-276}$, Fig. 1A). This gave a

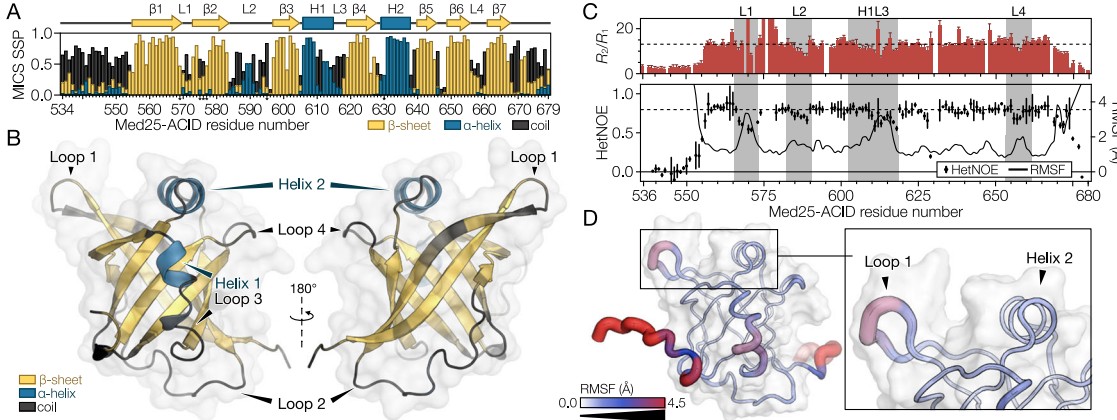

**Fig. 2 | Characterization of the *At*Med25-ACID domain structure using NMR, MD, and AlphaFold2. A** Secondary structure propensities from assigned chemical shifts calculated using MICS. Asterisks indicate residues for which only one or fewer [13]C chemical shifts could be assigned. **B** AF2 structure prediction of Med25-ACID colored using the SSP scores in panel **A. C** Analysis of Med25-ACID fast timescale dynamics using NMR relaxation parameters, {[1]H}–[15]N hetNOEs and MD simulation. Regions showing increased dynamics, indicated by lowered $R_2/R_1$ or lowered

hetNOEs or by increased root-mean-square fluctuations (RMSF), are highlighted in gray. $R_2/R_1$ error bars represent propagated standard errors. HetNOE errors are standard deviations of three technical replicates. Dashed horizontal lines indicate average values of $R_2/R_1$ and hetNOEs for the folded part of the ACID domain. **D** Mapping of MD derived residue fluctuations onto the Med25-ACID structure. Source data are provided as a Source Data file.

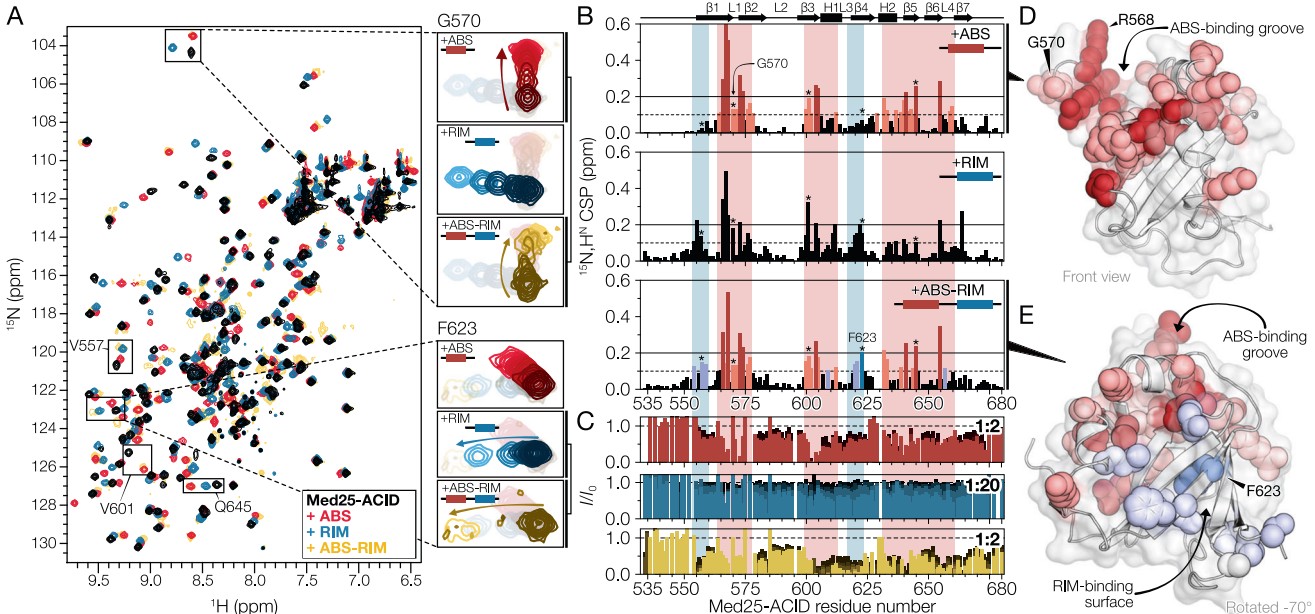

**Fig. 3 | Mapping of DREB2A binding sites in Med25-ACID using NMR. A** [15]N-HSQC spectra of [15]N labeled Med25-ACID in complex with ABS (red), RIM (blue). and ABS-RIM (yellow). Zoom: residues affected differently depending on the DREB2A fragment. **B** [15]N,H[N] CSPs of Med25-ACID induced by ABS (top), RIM (middle), and ABS-RIM (bottom). Red and blue background: regions forming the ABS and the RIM binding sites, respectively. Bars are colored according to their residue color in panels **D** and **E**. *Residues highlighted in panel **A**. **C** Same as **B**, but tracing HSQC

relative peak intensities as a function of DREB2A fragment concentration (dark to light). Molar ratios for the final intensity level (light color) are provided in the top right corner. **D** Mapping of ABS induced Med25-ACID CSPs on the AF2 structure. **E** Mapping of Med25-ACID CSPs induced by the bivalent ABS-RIM fragment. Blue spheres: RIM induced additional CSPs. **D** and **E** are colored according to significance levels of 0.1 (light) and 0.2 ppm (dark). Source data are provided as a Source Data file.

$K_d$ of $3.1 \pm 0.7\,\mu\text{M}$, more than five-fold higher than that of the wild type (Table 1, Supplementary Fig. 7), confirming that L1 is important for binding. Thus, both NMR and thermodynamics supported the importance of the groove for the interaction with DREB2A.

Since inclusion of the RIM resulted in a higher affinity (Table 1), we performed an NMR titration using the bivalent ABS-RIM to determine the interaction site of the RIM. Similar CSPs, both in magnitude and direction, were observed for residues in the binding groove, suggesting similar binding of the ABS with and without the RIM. However, additional CSPs were observed on a surface of Med25-ACID extending

from the ABS-binding groove (Fig. 3E). Furthermore, the addition of ABS-RIM resulted in a reduction of the spectral quality with many peaks disappearing at stoichiometric and higher DREB2A concentrations. Analysis of NMR peak intensity changes identified several regions affected by addition of ABS-RIM (Fig. 3C) and mapping the largest effects (Supplementary Fig. 6B) highlighted the same area as the additional CSPs (Fig. 3E), further supporting this region as the Med25-ACID binding surface of the RIM.

While ITC experiments using the RIM fragment (DREB2A[255-276], Fig. 1A) did not show any binding to Med25-ACID by itself, NMR

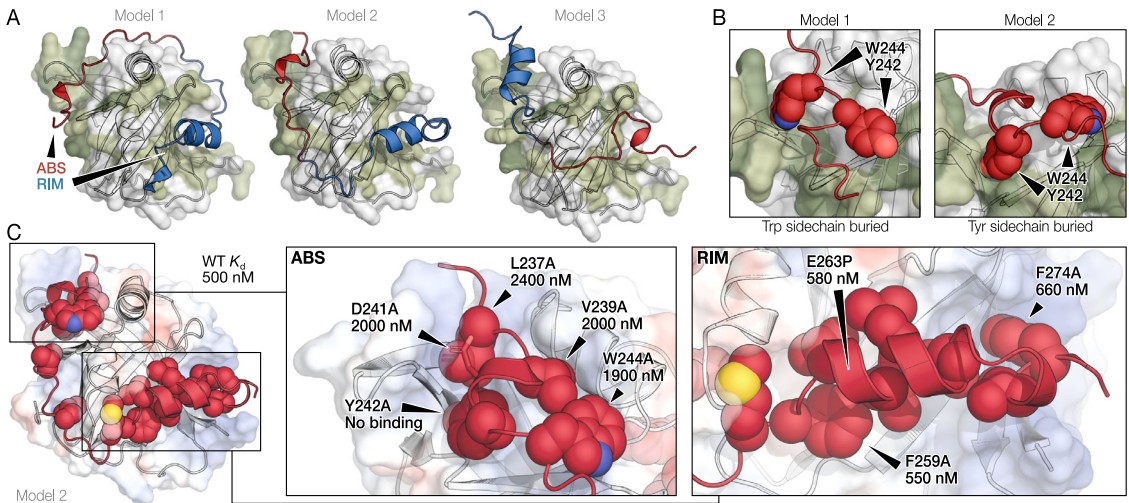

**Fig. 4 | Modeling of the Med25-ACID:DREB2A complex using AF2. A** Three structure models of Med25-ACID:DREB2A predicted using AF2 shown with the ABS-RIM induced Med25-ACID CSPs (green). The ABS (red) was predicted to form a single turn helix while the RIM (blue) was predicted to form a 10-residue helix.

**B** Comparison of the packing of Tyr242 and Trp244 in model 1 (left) and 2 (right). **C** AF2 complex model 2 with the hydrophobic residues shown as spheres and the affinity effect of specific DREB2A variants shown. Source data are provided as a Source Data file.

titrations showed large CSPs upon its addition. Unexpectedly, both the binding groove and the binding surface of Med25-ACID were affected, suggesting an ability of the RIM to bind both sites (Fig. 3B). Analysis of the RIM-induced CSPs as a function of concentration identified two different affinity regimes, with a $K_d$ of 200-400 μM for the groove and a $K_d$ of 3−6 mM for the surface (Supplementary Fig. 6C). While this indicated that the RIM interaction with the binding surface was relatively weak compared to that of the groove, the RIM interaction of the bivalent DREB2A fragments will be influenced by the effective concentration imposed by high-affinity ABS binding. Extrapolation to a linker length of 13 residues, as in the bivalent ABS-RIM, gave an effective RIM concentration of around 9 mM near Med25-ACID with the ABS bound[32], resulting in an expected $65 \pm 10\%$ saturation of the RIM towards the binding surface. Comparison of ABS, ABS-RIM, and RIM bound state chemical shifts showed that residues located in the ABS-binding groove were affected differently by the RIM, compared to the ABS and ABS-RIM fragments, which showed similar CSPs in the groove, while residues of the RIM-binding surface showed similar perturbations for the RIM and ABS-RIM fragments (Fig. 3A). This suggested that, for their cognate sites, the bound state of the isolated SLiMs resembles that seen in the bivalent fragment, but that using the isolated motifs resulted in incorrect binding modes. Together, these results suggested that the ABS binds the groove, while the RIM targets the binding surface.

### AF2 modeling supports bivalent binding

We next exploited the AF2 heterodimer modeling option on the ColabFold server[33] to predict the structure of the Med25-ACID:ABS-RIM complex. Out of three different complex structure configurations, two predicted the ABS to bind the groove and the RIM to bind the surface, in agreement with the NMR data (Fig. 3D, E). The third model (model 3) swapped the positions of the ABS and RIM and was therefore discarded based on the NMR data (Fig. 3A). The two remaining models differed by the direction of DREB2A wrapping around the ACID domain, with model 1 having DREB2A wrapping along β2, while model 2 had DREB2A wrapping around the H1L3 side. In addition, model 1 had Trp244 buried in the binding groove, while model 2 had Tyr242 buried in the groove (Fig. 4B). Based on the mutational analysis, which showed Tyr242 to be crucial for binding while Trp244 was less important (Table 1), model 2 was considered the more likely model.

The model indicated that Leu237 and Val239 are important for binding as they were buried in the binding groove. To test this, two mutant DREB2A fragments were produced, individually substituting each residue with alanine, and the affinities were measured using ITC (Table 1). The V239A and L237A variants both exhibited slightly reduced affinities with $K_d$ values of $2.0 \pm 0.2\,\mu M$ and $2.4 \pm 0.3\,\mu M$, respectively. However, the L237A variant exhibited distinct thermodynamics, featuring a favorable entropic contribution. This indicated structural effects of mutations of ABS residues buried in the binding groove, consistent with the ITC results of the Y242A variant. Together the ITC and NMR (Fig. 2A) data support model 2, although alternative configurations may be populated under certain conditions.

Based on the AF2 model, binding of the RIM involves a hydrophobic surface of Med25-ACID (Fig. 4C), which was in line with the limited effect of mutating single phenylalanine residues (Table 1). All AF2 models demonstrated α-helical structure in the RIM, yet a helix-disrupting DREB2A E263P variant showed unchanged affinity ($K_d = 580 \pm 40\,nM$; Table 1), indicating that helix formation is not a prerequisite for binding and that multiple binding modes may exist. In agreement with the H1L3 wrapping model (Fig. 4A, model 2), evolutionary analysis of plant Med25-ACID revealed that the opposite side, in particular β2, was less conserved than the rest of the ACID domain (Supplementary Fig. 8), suggesting reduced requirements to support specific interactions. Overall, the complexes predicted using AF2 agreed well with the experimental data (Fig. 4).

### Multiple bound states of DREB2A uncovered using CEST

Previous characterization of free DREB2A revealed that the region around Pro245 exists in two states, likely resulting from Trp244 stabilizing the *cis* peptidyl-prolyl Trp244-Pro245 isomer[34]. Quantification of HSQC peak volumes of free DREB2A (Supplementary Fig. 9 and Supplementary Table 1) showed that $30 \pm 1\%$ was in the *cis* proline form, resulting in an equilibrium constant ($K_{eq,trans \rightarrow cis}$) of $0.42 \pm 0.02$. Attempts to use NMR to study DREB2A in the Med25-ACID bound state were hindered by line broadening. Instead, we exploited the CEST-type NMR experiments[34,35], which can detect lowly populated states by observing the modulation of the unbound state. Initial analysis of all non-overlapping peaks (Supplementary Fig. 10A) revealed two-state behavior of both the *cis* and *trans* proline isomers of the ABS region, indicating that both isomers bind Med25-ACID (Fig. 5A), although differences in the bound populations suggested different affinities of

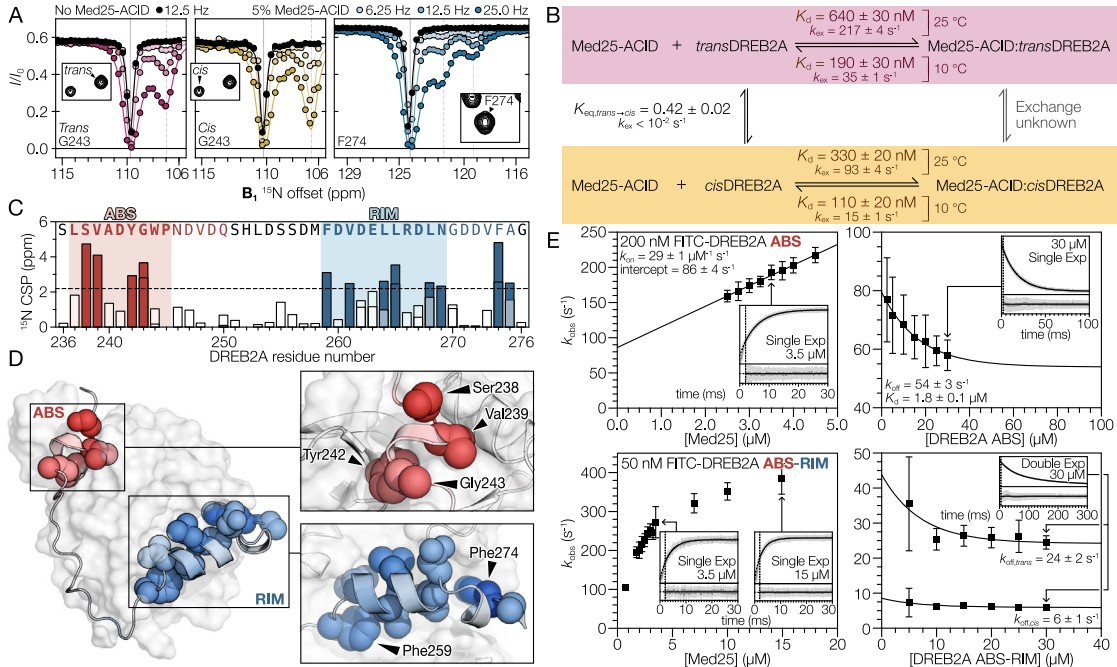

**Fig. 5 | CEST and stopped-flow spectroscopy analyses of DREB2A showing two bound states. A** Example $^{15}N$-CEST profiles for DREB2A Gly243 in the *trans* and *cis* proline states, illustrating binding of both isomers, and Phe274 illustrating three-state behavior of sequentially distant residues. Shown profiles were recorded with (colored) and without (black) 5% unlabeled Med25-ACID at 25 °C. Vertical gray lines indicate the chemical shifts of detected populations, with the solid line marking the free state and dashed line(s) marking the bound state(s). **B** Kinetic model for analysis of the $^{15}N$-CEST data with derived affinities. Exchange between the *cis* and *trans* bound states is not considered. **C** DREB2A $^{15}N$ CSPs extracted from the CEST experiment. The dashed line indicates an arbitrary significance level of 2.2 ppm. **D** Mapping of DREB2A $^{15}N$ CSP on the AF2 complex model. Residues with significant CSPs are shown. **E** Stopped-flow fluorescence spectroscopy analysis of ABS (top) and ABS-RIM (bottom) to determine association (left) and dissociation (right) rate constants. Data points and error bars represent mean and standard deviation of 19 or more individually fitted traces assuming a normal distribution. Example averaged traces are shown for each analysis. Averaged traces and derived $k_{obs}$ for all data points are shown in Supplementary Figs. 11 and 12. Source data are provided as a Source Data file.

*cis* and *trans* DREB2A towards Med25-ACID (Supplementary Fig. 10B). Many peaks from the RIM displayed three-state behavior (Fig. 5A) with large CSP differences between the two bound states, suggesting structural heterogeneity. The three-state behavior may be caused by different structural constraints imposed by the two proline isomers. We therefore produced a DREB2A variant with a substitution of Pro245 with alanine (ABS-RIM$_{P245A}$), which eliminated the three-state behavior of the RIM (Supplementary Fig. 10C). Structural heterogeneity of the bound RIM, driven by Pro245 isomerization, may explain the HSQC peak intensity loss of Med25-ACID RIM-binding surface residues upon binding of ABS-RIM (Fig. 3C and Supplementary Fig. 6B).

To enable global analysis of the CEST data, we grouped the CEST profiles into *trans* ABS, *cis* ABS, and apparent three-state RIM peaks (Supplementary Figs. 10A–C). Analysis of the ABS confirmed the distinct behavior of *cis* and *trans* isomer peaks, with the *cis* isomer having a larger bound fraction of $6.8 \pm 0.1\%$ compared to the $3.6 \pm 0.1\%$ of the *trans* isomer, supporting a difference in affinity. Based on the overall affinity of $500 \pm 20$ nM, determined using ITC, we estimated the $K_d$ of the *trans* and *cis* isomers to $640 \pm 30$ nM and $330 \pm 20$ nM, respectively (Fig. 5B and Supplementary Method 1). The differences in bound fractions were accompanied by different exchange rates, indicating that the dissociation rate constants differed for the two isomers, similar to the proline isomerization effects described for the ACTR:NCBD interaction[36]. Global analysis of three-state ABS-RIM CEST profiles suggested that approximately 4.7% was in one of the two bound states, indicating that the RIM of the bivalent DREB2A fragment was found exclusively in a bound state (Supplementary Method 1). This was more than expected based on the NMR-derived affinity of the RIM alone and the linker-length-based estimates of the effective concentration, suggesting either that the linker also contributes to binding

or that binding of ABS results in cooperative binding of the RIM, as observed for other Med25-ACID interactions[37].

We then extracted $^{15}N$ CSPs for ABS-RIM and mapped them onto the AF2 complex model (Fig. 5C, D). Large CSPs were observed for both motifs while the linker region showed only minimal perturbations, indicating weak contact and binding effects. In agreement with the complex model, hydrophobic residues in the interface (Fig. 4C) generally experienced larger CSPs (Fig. 5C, D).

To further explore the observed proline-dependent effect on the exchange rate, we determined the association ($k_{on}$) and dissociation ($k_{off}$) rate constants with stopped-flow fluorescence spectroscopy using FITC-labeled DREB2A and unlabeled Med25-ACID (Fig. 5E and Supplementary Fig. 11). A $k_{on}$ of $29 \pm 1\ \mu M^{-1}\ s^{-1}$ was obtained for the ABS; however, ABS-RIM showed a non-linear concentration dependence of the observed rate constant ($k_{obs}$), indicating a complex binding model with potential contributions from non-specific binding. Consequently, a $k_{on}$ could not be obtained for the ABS-RIM. The dissociation was analyzed using competitive displacement experiments. Time-dependent fluorescence traces of the ABS dissociation could be adequately fitted using a single exponential decay function, whereas a double exponential function was necessary to obtain a good fit of the ABS-RIM dissociation (Fig. 5E and Supplementary Fig. 12). This was in accordance with the two distinct $k_{ex}$ values obtained from the CEST analysis (Fig. 5B) and suggested that the effect of proline isomerization was insignificant in the absence of the RIM. The $k_{off}$ values obtained for the ABS-RIM (Fig. 5E) were similar to the *cis* and *trans* proline state $k_{ex}$ values obtained from CEST experiments conducted at 10 °C (Fig. 5B), confirming that the biphasic behavior of the dissociation kinetics was the result of proline isomer dependent structural differences.

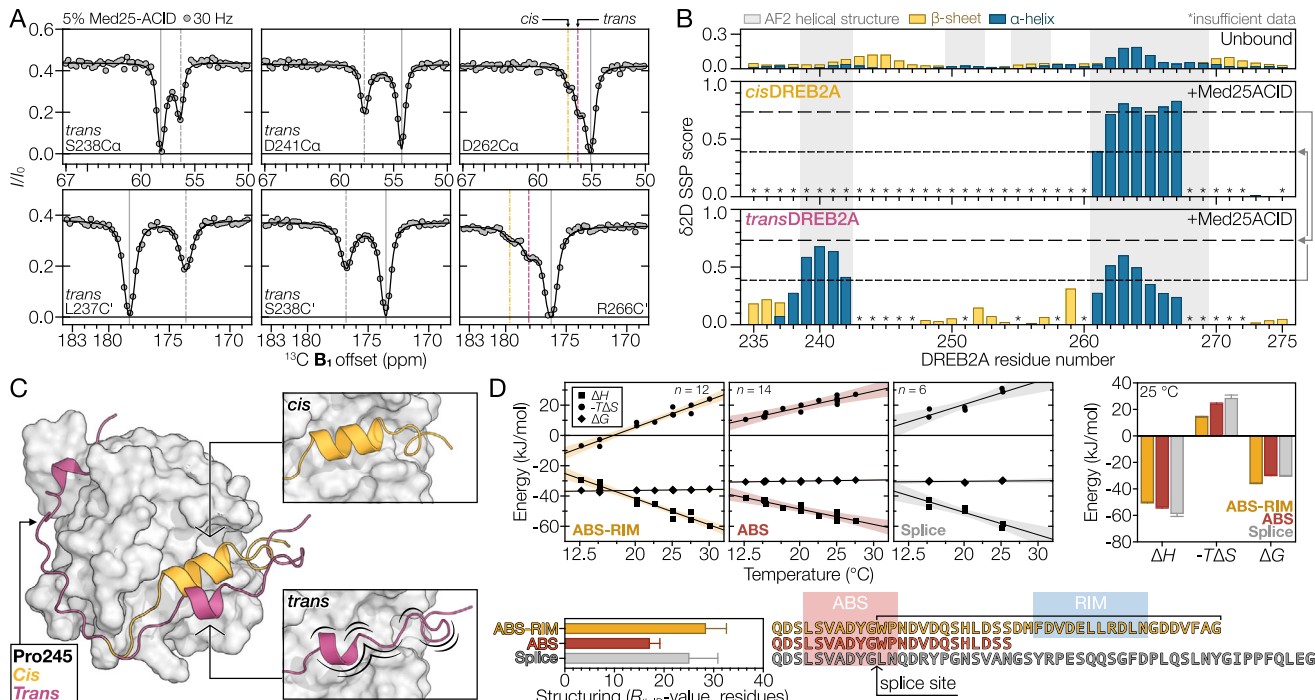

**Fig. 6 | $^{13}$C-CEST and thermodynamic analyses show coupled binding and folding of DREB2A. A** $^{13}$C-CEST profiles of DREB2A with 5% Med25-ACID. **B** SSPs calculated using δ2D of DREB2A in the unbound and the Med25-ACID bound states. Bound state SSPs were predicted for chemical shifts extracted from the minimum and maximum perturbations using $^{15}$N, $^{13}$C$^α$, and $^{13}$C′. Unbound DREB2A chemical shifts were extracted from previously published data. *Data insufficient to predict SSP. **C** Model illustrating the difference in bound state secondary structure observed for the *cis* and *trans* proline states. **D** Thermodynamic characterization of the DREB2A interaction with Med25-ACID. (Top) temperature dependence of the thermodynamic parameters and derived parameters at 25 °C. Error bands for the linear fits are 95 % confidence intervals and error bars for the derived parameters obtained from the standard errors of the linear fits. The *n* number is the number of experiments for each fragment. (Bottom) $R_{th,ID}$ based on the ID-adapted Spolar-Record method[22] and alignment of fragment sequences. $R_{th,ID}$ error bars were obtained using a Monte Carlo approach incorporating standard errors from the linear temperature dependence of the thermodynamic parameters. Source data are provided as a Source Data file.

## DREB2A undergoes coupled folding and binding forming two short α-helices

To analyze the secondary structure of the DREB2A bound state, we determined the $^{13}$C$^α$ and $^{13}$C′ chemical shifts using triple-resonance $^{13}$C-CEST experiments[38,39]. Due to the reduced sensitivity of the experiments, it was not possible to analyze the relatively weak *cis* proline-associated peaks. However, the $^{13}$C-CEST profiles of the RIM (Fig. 6A and Supplementary Fig. 13) supported the three-state behavior seen in the $^{15}$N-CEST profiles (Fig. 5A), not only confirming the existence of two bound states, but also indicating that the structure of the two bound states differed. CEST-derived CSPs of the DREB2A$_{P245A}$ variant showed that whereas the ABS was not greatly affected by the proline substitution, the RIM displayed reduced CSPs, resembling the less affected of the two bound states seen in the WT (Supplementary Figs. 10C and 14). This suggested that the large RIM CSPs seen in the WT belong to the *cis* proline bound state. To quantify the secondary structure of the two bound states, we compiled two sets of $^{15}$N, $^{13}$C$^α$, and $^{13}$C′ chemical shifts, extracted from the minimum (*trans*) and maximum (*cis*) CEST CSPs and submitted these to the δ2D SSP webserver[40] (Fig. 6B). For the *cis* state, predictions indicated formation of well-folded helical structure in the RIM, matching the AF2 complex model (Fig. 6C). Conversely, the *trans* state showed reduced helical structure in the RIM, suggesting that it can bind to Med25-ACID in both a helical and a non-helical structure, depending on the isomeric state of Pro245.

To further explore the degree of structuring, we estimated the number of residues folding upon binding ($R_{th,ID}$) using the ID-adapted thermodynamic Spolar-Record method[22,41]. For this, we used three DREB2A fragments; the monovalent ABS fragment, the bivalent ABS-RIM fragment, and the splice variant fragment (Fig. 6D). The $R_{th,ID}$ value of the ABS (17 ± 3) was larger than the number of residues in the DREB2A fragment which suggested structuring of the ACID domain. Comparison of the $R_{th,ID}$ values of ABS and ABS-RIM (28 ± 5) suggested that the RIM undergoes coupled folding and binding, matching the δ2D chemical shift analysis (Fig. 6B). The reduced entropic penalty of the longer bivalent fragment suggested dehydration of hydrophobic surface upon binding of the RIM. In agreement with this, substitution of large hydrophobic sidechains with alanine resulted in an increased entropic penalty, which was compensated by favorable enthalpy (Table 1), suggestive of a highly adaptable interface enabled by enthalpy-entropy compensation. Binding of the splice variant, lacking the RIM, but containing long flanking regions, also resulted in folding of more residues (26 ± 7) than for the ABS fragment. In this case, thermodynamic parameters suggest some association of the C-terminal context with the unfavorable change in entropy possibly due to conformational restrictions of ABS residues. Thus, as for the RCD1:DREB2A interaction[22], the motif context ensures coupled folding and binding through intricate enthalpy-entropy compensation.

## Discussion

In this study, we explored how a co-regulator interacts with a TF through a disordered functional hot spot. Focusing on the DREB2A interaction with Med25 enabled comparison to previous studies with the negative regulator RCD1 and the co-activator TAF4, both of which bind DREB2A through their RST domain[42]. DREB2A interacts with Med25-ACID using a 40-residue region containing two SLiMs. The primary Med25-ACID binding SLiM, the ABS, is characterized by key aromatic, acidic and proline residues, but did not correlate with

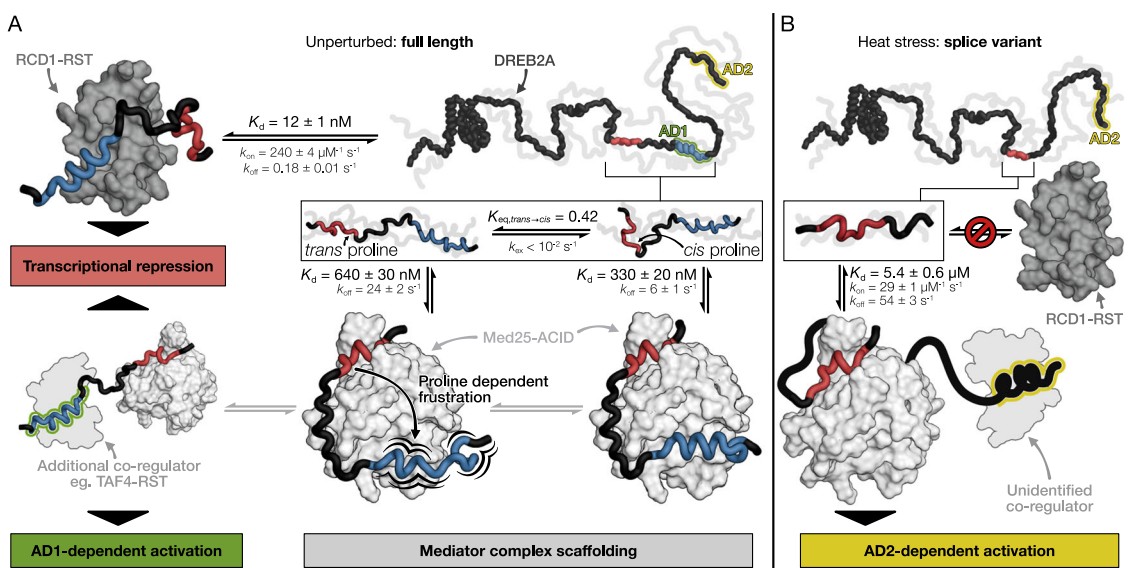

**Fig. 7 | Stress dependent positive and negative regulation of DREB2A. A** Bivalent DREB2A interaction region, produced under non-stress conditions, forms a complex with Med25-ACID using both the ABS and the RIM, whereas the high-affinity RCD1-RST interaction is driven by the RIM. The ABS interaction with Med25 scaffolds the mediator complex while proline-dependent frustration of the RIM may promote binding of additional co-regulators. **B** Heat stress induced synthesis of a DREB2A splice variant lacking the RIM (and AD1) is unable to bind RCD1-RST but retains the ability to scaffold mediator through the ABS and activate transcription through AD2. Affinities are given at 25 °C while kinetic rate constants are given at 10 °C.

transcriptional activity in yeast (Fig. 1A and Supplementary Fig. 3B) and plants[25]. The other SLiM, the RIM, constitutes a secondary binding motif for Med25-ACID, but is the primary interaction motif for the RST domains of RCD1 and TAF4 (Fig. 7A)[21], and overlaps with AD1. Thus, the interaction regions of co-regulator and repressor overlap, but each interaction is driven by distinct motifs with both interactions showing a 10-fold higher affinity in the bivalent configuration (Table 1)[22].

We also demonstrated that a DREB2A splice variant, expressed under stress and lacking most of the bivalent interaction region including the RIM and AD1, retained the ability to bind Med25-ACID. This suggests that the Med25 interaction may serve as a scaffold for the transcriptional machinery, allowing for DREB2A transcriptional activation through either AD1 or AD2. Although DREB2A has a higher affinity for RCD1-RST ($K_d = 12 \pm 2$ nM)[22] than for Med25 ($520 \pm 20$ nM, Table 1) the in vivo competition scenario between e.g. RCD1, TAF4, and Med25 for DREB2A will depend on many other factors, including concentrations. However, alternative splicing of DREB2A will shortcut such regulatory fine-tuning of competition by removing the RIM, thereby eliminating RCD1 as a repressor while leaving AD2 available (Fig. 7B). This provides an obvious mechanism for functional switching in a transcriptional pathway.

As a hub protein, RCD1 binds many stress-related TFs with low nanomolar affinity[20] and could act as a leaky expression repressor of e.g. DREB2A. Stress responses may involve rapid degradation of RCD1[21], thus releasing bound DREB2A to interact with Med25 using its bivalent high-affinity region. This would provide a strong initial response, while the prolonged response would be of a lower potency due to stress-induced expression of the splice variant[21] lacking the ABS context.

Biophysical and structural characterization together with AF2 modeling uncovered a groove and a surface on *At*Med25-ACID capable of binding the ABS and RIM, respectively, and CEST NMR confirmed involvement of both motifs in binding. Although the RIM-binding surface (Fig. 3E) corresponds to the *h*Med25-ACID:VP16-H2 interaction interface, the ABS-binding groove (Fig. 3D) does not match the VP16-H1 binding site[10]. Similarly, a RIM binding surface equivalent was reported for p53-TAD1[11] and ERM[12]. However, neither ERM nor p53-TAD2 bound to a groove corresponding to that identified for the ABS

and although a 40-residue ETV4 fragment induced CSPs in L1, the effect was generally dispersed across the ACID domain[8] (Supplementary Fig. 15). Thus, the *At*Med25-ACID:DREB2A interaction contributes to the diversity in Med25:TF complexes and highlights the theme of multivalency in Med25-ACID:TF interactions[8–11] conserved across kingdoms. The most striking observation is the identification of the ABS-binding groove. Although this appears unique for the interaction between Med25-ACID and DREB2A, this may be due to experimental challenges of NMR studies using bivalent interaction partners and could be a general feature. Furthermore, the NMR titrations employing the individual motifs and the bivalent association kinetics suggested that alternative binding modes may be populated under certain conditions depending on the cellular context.

Structural heterogeneity and dynamics characterize mediator:TF interactions[43,44], making structural analysis challenging. Here we show that DREB2A exists in highly populated *cis* and *trans* Pro245 states. According to CEST analysis, both isomers bound Med25-ACID, with the RIM showing two distinct bound states, displaying considerable α-helical propensity in the *cis* bound state. This both supported the AF2 complex model and the Spolar-Record analysis by confirming the presence of helical structure in the RIM, and explained why the helix-breaking E263P mutation had no significant effect on affinity. The structural heterogeneity is unlikely to be caused by proline-dependent conformational changes in the ACID domain, but rather by restraints imposed on the disordered DREB2A region linking the ABS and the RIM. This is also in accordance with the kinetic analysis revealing no effects of proline isomerization in the interaction of the isolated ABS. Thus, our characterization of the *At*Med25-ACID:DREB2A complex demonstrates that proline isomerization contributes to defining the mediator:TF interactome by providing the characteristic heterogeneity. Proline isomerization is an emerging property in ID-based protein-protein interactions[36,45]. Thus, proline isomerization has recently been shown to enable differential binding of transcription factors to the NCBD αα-hub domain of CBP[36,46] and to modulate the liquid-liquid phase separation of IDPs[47]. Proline isomerization can also function as a molecular timer controlling the amplitude and duration of cellular processes[45]. Molecular timing appears highly relevant for DREB2A in regulation of stress-responses through its molecular switch

region binding transcriptional activators and repressors. Whether the DREB2A proline is a substrate for one of the stress-associated *Arabidopsis* cyclophilins *cis-trans* isomerases[48] remains an open question.

Interactions in structurally heterogeneous complexes may have a high degree of energetic frustration[49,50], which is also likely for the bivalent interaction of DREB2A with Med25-ACID (Fig. 7A). Bivalency allows high-affinity binding and thereby specificity in interactions. However, for the DREB2A:Med25-ACID interaction, the weak secondary binding site and energetic frustration, particularly of the *trans* bound state, may allow other co-regulators, such as TAF4[42] and RCD1[20], to bind the RIM and either displace DREB2A from Med25 or participate in ternary complexes (Fig. 7A). ID-based molecular competition has been demonstrated for the HIF1α and CITED2 transcription factors in their binding to the TAZ1 domain of CBP as part of the hypoxic response[51]. In summary, the results suggests mechanisms for the complex interplay between co-regulators and transcription factors, highlighting the importance of SLiM context, proline isomerization, and alternative splicing in molecular switching enabling fine-tuning of transcriptional regulation.

# Methods

## Protein expression and purification
All protein fragments were produced with an N-terminal Glutathione-S-transferase (GST) tag which was removed using Tobacco Etch Virus (TEV) protease and expressed from *E.coli* BL21 DE3 cells. DREB2A (AT5G05410) variants were expressed at 37 °C for three hours. The DREB2A splice variant (AT5G05410.2) was based on GenBank ID CD530293[21]. Due to stability and aggregation considerations, Med25-ACID (AT1G25540) was expressed with additional N-terminal residues[52]. Expression of Med25-ACID$_{532-680}$ was induced at high OD (0.8-1.0) and cultured overnight at 18 °C. Cells were resuspended in 20 mM Tris-HCl pH 8.0, 100 mM NaCl lysis buffer and sonicated. The lysate was incubated with glutathione Sepharose for one hour and washed using the lysis buffer. Aliquots of 5 mL resin were resuspended in 20 mL of lysis buffer with 5 mM dithiothreitol (DTT) and 1 mM EDTA and the protein was cleaved overnight at 4 °C using TEV protease. For DREB2A, cleaved protein was freeze-dried and buffer-changed to experimental buffer using size exclusion chromatography. Med25-ACID was further purified using cation exchange (Source 15 S) at pH 8.5 and the buffer was changed using size exclusion chromatography (Superdex S75 10/300 GL). Isotope-labeled proteins were produced by growing the initial culture in lysogenic broth and transferring to half volume M9 minimal media 30 min before induction.

## Isothermal titration calorimetry
ITC was performed on either a MicroCal ITC200 or a Malvern Panalytical PEAQ-ITC. All ITC experiments were performed using a 50 mM HEPES pH 7.4, 100 mM NaCl and 0.5 mM tris(2-carboxyethyl) phosphine (TCEP) buffer. All experiments were performed using a ~1:10 cell to syringe protein concentration ratio. For high-affinity DREB2A fragments, the cell concentration was around 15 μM, while lower-affinity interaction experiments used higher concentrations. Samples were degassed by centrifuging at 20,000 x *g* for 20 minutes at experimental temperature. For three DREB2A fragments, experiments were performed over a range of temperatures to determine the $\Delta C_p$. Reported thermodynamic parameters are given at 25 °C. Data were analyzed using Origin and inhouse scripts using the one-set-of-sites model as documented by MicroCal and Origin. Dilution heats were fitted using an offset parameter. Most experiments were performed using a 0.5 μl initial injection, which was discarded in the analysis, followed by 18 injections of 2.0 μl.

## High-throughput screening for ADs in *Arabidopsis* TFs
The DREB2A protein sequence was broken up into forty amino acid tiles with a step size of ten amino acids. Yeast codon optimized

sequences for each DREB2A tile were synthesized and cloned in bulk into pMVS1421[28] to create a synthetic TF consisting of an N-terminal mCherry tag, a mouse Zif269 DNA-binding domain, an estrogen binding domain, and the tested DREB2A tile expressed under the yeast ACT1 promoter. The expression cassette was cloned using homologous recombination into the mating type A strain DHY211 (courtesy of Angela Chu and Joe Horecka) at the *URA3* locus as previously described[28]. Positive strains carrying the synthetic transcription factor were mated to yeast carrying a GFP reporter driven under the P3 promoter as previously described[28]. Yeast were sorted based on reporter to transcription factor ratio (GFP:mCherry), and each bin was sequenced to determine the abundance of each fragment tested. Reads were aligned and quantified using BWA aligner and SAMtools. Fragments counts were analyzed using custom scripts in python to generate the AD score. AD score was calculated by taking the dot product of fragment abundance in each bin by the median GFP:mCherry ratio of each bin and then Z-score normalized, as previously described[28]. Activity of synthetic control TFs with known ADs is shown in Supplementary Fig. 3.

## Bioinformatics
*At*DREB2A orthologous sequences were obtained from the PLAZA database[53] and aligned using CLUSTAL Omega[54,55]. The MEME suite[56] was then used to define a motif, which described the binding region. ID-profiles were predicted using DISOPRED[57], and α-helix propensity was analyzed using Agadir[58].

## Small-angle X-ray scattering
SAXS data were acquired on the PETRA III P12 beamline using in-line size exclusion chromatography. A single sample of 5.8 mg/mL Med25-ACID$_{532-680}$ was prepared in 20 mM Na$_2$HPO$_4$/NaH$_2$PO$_4$ pH 6.5, 100 mM NaCl and 1 mM TCEP. Scattering data was collected from the single peak on the elution profile. Buffer reference scattering data was collected from areas surrounding the protein elution peak. Scattering data were recorded and processed by staff at the facility in Hamburg and the data were analyzed using the ensemble optimization method[59]. Scattering data were analyzed using the ensemble optimization method (EOM)[60] with a structure pool extracted from a 1.7 μs molecular dynamics simulation of the AF2 Med25-ACID$_{540-680}$ structure. Missing N-terminal residues were generated using the EOM RANCH algorithm.

## Molecular dynamics simulation
Molecular dynamics simulations were performed using GROMACS[61] using single-precision floating point calculations. The MD simulations were initiated from the equilibrated AF2 model of Med25-ACID$_{540-680}$ using the AMBER99SB forcefield[62] and TIP3P water model[63]. Overall system charge was naturalized by adding Na$^+$ and Cl$^-$ ions with additional ions added for a final concentration of 100 mM NaCl.

## NMR spectroscopy
NMR data were acquired on Bruker AVANCE 600, 750 or 800 MHz ($^1$H) spectrometers equipped with cryogenic probes. Free induction decays were transformed and visualized using NMRPipe[64] and analyzed using the CcpNmr Analysis software[65]. All NMR samples were prepared using 10% (v/v) D$_2$O, 0.02% (w/v) NaN$_3$, 0.2 mM 4,4-dimethyl-4-silapentane-1-sulfonic acid (DSS), 20 mM Na$_2$HPO$_4$/NaH$_2$PO$_4$, and 100 mM NaCl. All non-CEST samples were prepared at pH 6.5. Backbone assignment of Med25-ACID$_{532-680}$ in the unbound state was done manually from analysis of $^{15}$N-HSQC, HNCACB, HNCOCACB, HNCO and HNCACO experiments (BMRB ID: 52040). Assignment of the DREB2A$_{234-256}$ bound state (BMRB ID: 52042) was done using BEST[66] TROSY (BT)-HSQC, BT-HNCA and $^{13}$C$^β$ optimized BT-HNCACB experiments[67]. All Med25-ACID assignment samples were recorded using 500 μM $^{13}$C,$^{15}$N-labeled Med25-ACID$_{532-680}$ and, if relevant, 2 mM unlabeled DREB2A$_{234-256}$, Assignment of the unbound DREB2A$_{234-276}$ fragment

was acquired from previous work (BMRB ID: 51055)[22]. SCSs were calculated using the SBiNLab[68] and PONTECI[69] web tools and are reported as mean, with error bars representing the standard deviation. {$^1$H}-$^{15}$N hetNOEs were recorded in triplicates, processed individually and reported as mean and standard deviation. Unbound DREB2A$_{234-276}$ *cis*- and *trans*-proline populations were quantified from HSQC spectra recorded with a recycle delay of 6 seconds by fitting peak volumes using the PINT software[70]. Five non-overlapping peaks, visible in both isomeric states, were quantified and the populations were estimated as the volume of a given peak divided by sum of the *cis* and *trans* peak volumes.

## NMR titration analyses

All NMR titration data were acquired on a Bruker AVANCE III HD 600 spectrometer using between 100 and 150 μM $^{15}$N-labeled Med25-ACID$_{532-680}$ (BMRB ID: 52041, 52042, 52043). The binding was monitored with $^{15}$N-HSQC spectra using constant Med25-ACID concentrations with increasing concentration of DREB2A ligand. All titration series included a spectrum with no ligand, which was used as the unbound state when calculating CSPs. $^{15}$N,H$^N$ CSPs were calculated as the Euclidian distance using a 0.154 weight for the $^{15}$N chemical shift[71].

## Chemical exchange saturation transfer experiments

All CEST NMR data was acquired on a Bruker AVANCE Neo 800 spectrometer with a 5 mm CPTXO Cryoprobe. CEST samples were prepared at pH 6.1 using 500 μM $^{13}$C,$^{15}$N -labeled DREB2A$_{234-276}$, or 1 mM DREB2A$_{234-276,P245A}$, with, if relevant, approximately 5% molar ratio of unlabeled Med25-ACID$_{532-680}$. $^{15}$N-CEST data was acquired using previously published pulse sequences[34] using three different **B$_1$** field strengths. $^{15}$N-CEST data acquired at 25 °C was done using 25, 12.5, and 6.25 Hz **B$_1$** fields while experiments at 10 °C were done using 40, 20, and 10 Hz **B$_1$** fields. $^{13}$C-CEST data was acquired using pulse sequences[38,39] provided by prof. Lewis Kay with a **B$_1$** field of 30 Hz. For all CEST data, free induction decays were transformed and visualized using NMRPipe[64] and peak intensities, $I$, were quantified as the height at the specific peak position. Overlapping peaks and peaks showing no exchange were discarded in global fitting procedures, but approximate CSP were extracted when possible. Analysis and fitting were done using ChemEx[72]. *Cis*- and *trans*-proline peaks were analyzed separately using a two-state model and C-terminal residues, which displayed three-state behavior, were analyzed using a three-state model. Peaks that could not be grouped were analyzed individually to determine CSPs. $^{15}$N-CEST global fitting groups are shown in Supplementary Fig. 10.

## AF2 complex modeling

AF2 modeling of the Med25-ACID:DREB2A complex was done using the ColabFold webserver[33] (accessed June 2022) by providing the sequences of isolated Med25-ACID domain and the DREB2A$_{234-276}$ fragment using the heterodimer syntax. The server reports five models for each prediction. Two predictions were performed, from which three different configurations were observed (Fig. 4A), although the DREB2A chain was generally assigned a low confidence (Supplementary Fig. 4D). Models were filtered based on binding site data and mutational analyses resulting in one model with satisfaction of experimental observations.

## Stopped-flow kinetics

Stopped-flow fluorescence experiments were performed using a sequential SX20 stopped-flow spectrometer (Applied Photophysics). DREB2A$_{234-276}$ was labeled using a fluorescein isothiocyanate (FITC) labeling kit (Pierce™) and purified with reversed phase chromatography (Zorbax 300SB C18, 70% ACN 0.1% TFA) to obtain single-labeled FITC-DREB2A. FITC-DREB2A$_{234-256}$ synthetic peptide was purchased from TAG Copenhagen A/S. Samples were prepared in 50 mM

HEPES buffer, pH 7.4, 100 mM NaCl, 1 mM DTT. FITC was excited at 490 nm and fluorescence detected at 519 nm, with a long pass cutoff filter at 515 nm. All samples were filtered through a 0.22 μm PVDF syringe filter prior to experiments and FITC samples were handled with aluminum foil to limit fluorophore damage. At least 20 fluorescence traces were obtained for each condition. Association kinetics were conducted under pseudo-first-order conditions by mixing of equal volumes of FITC-DREB2A (final concentrations of 50 and 250 nM for DREB2A$_{234-276}$ and DREB2A$_{234-256}$, respectively) and increasing concentrations of Med25-ACID$_{532-680}$. Dissociation rate constants were determined using competitive displacement experiments, mixing preformed Med25-ACID$_{532-680}$:FITC-DREB2A complex (1:0.1 μM for Med25-ACID:FITC-DREB2A$_{234-276}$ and 4:0.1 μM for Med25-ACID:FITC-DREB2A$_{234-256}$) with increasing concentrations of unlabeled DREB2A. Association traces of both DREB2A fragments were fitted using a single exponential function. Dissociation traces of DREB2A$_{234-256}$ were fitted using a single exponential decay function, while DREB2A$_{234-276}$ traces were fitted using a double exponential function. Traces were fitted individually and observed rate constants and standard deviations were obtained by fitting a cumulative Gaussian function to the distribution of rates. $k_{on}$ was obtained as the slope of the linear fit of the $k_{obs}$ concentration dependence. $k_{off}$ was determined from the exponential decay asymptote of the concentration dependence of the observed dissociation rate(s).

## Effective concentration calculation

The bound fraction of C-terminal motif of the bivalent DREB2A fragments was calculated using an empirical correlation between disordered linker length and effective concentration[32]. Extrapolating to the DREB2A motif linker length of 12.5 ± 2.5 residues resulted in an effective concentration of 9 ± 3 mM. Given the affinity of 4.7 ± 1.8 mM extracted from NMR titrations we calculated an expected saturation level of 65 ± 10%.

## Reporting summary

Further information on research design is available in the Nature Portfolio Reporting Summary linked to this article.

## Data availability

The bioinformatic, ITC, SAXS, stopped-flow and AF2 complex model data generated in this study have been deposited on GitHub [https://doi.org/10.5281/zenodo.10409674][73]. Processed NMR and MD data are also available on GitHub. The NMR chemical shift assignments for Med25-ACID in the free and DREB2A bound states have been deposited in the BioMagResBank (BMRB) under the following accession codes and DOIs: free Med25-ACID (52040 [https://doi.org/10.13018/BMR52040]), Med25-ACID with DREB2A ABS-RIM (52041 [https://doi.org/10.13018/BMR52041]), Med25-ACID with DREB2A ABS (52042 [https://doi.org/10.13018/BMR52042]), Med25-ACID with DREB2A RIM (52043 [https://doi.org/10.13018/BMR52043]). Source data for main and Supplementary items are provided with this paper. Source data are provided with this paper.

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

## Acknowledgements

This work was supported by the Novo Nordisk Foundation challenge grant REPIN, rethinking protein interactions (grant no.: NNF18OC0033926; to B.B.K. and K.S.) and by the Danish Research Council (grant no.: 9040-00164B to B.B.K. and K.S.). Activation domain assay was done under the National Science Foundation (NSF) (PGRP BIO-2112056 to L.C.S.; Postdoctoral Research Program IOS-1907098 to N.M.). NMR spectra were recorded at cOpenNMR, an infrastructure supported by the Novo Nordisk Foundation (grant no.: NNF18OC0032996). The authors would like to thank Dr. Anders Lønstrup Hansen for many important discussions regarding analysis and representation of data. We also thank Prof. Lewis E. Kay for sharing the $^{13}$C-CEST pulse sequences necessary for this work, Dr. Elise Delaforge for discussions and suggestions regarding analysis of CEST data and Stase Bielskute for suggesting the δ2D tool for analysis of IDP secondary structure. We wish to thank Cy Jeffries at the EMBL Hamburg beamline for recording, processing and providing initial analysis of SAXS data. We thank Profs. Ben Schuler and Per Jemth for valuable discussions of kinetics data leading to further exploration.

## Author contributions

F.F.T., A.P., S.E., Y.H.A.L., N.M., L.C.S, C.O.S, K.T., B.B.K., and K.S. designed the experiments. F.F.T., S.E., Y.H.A.L., N.M., and C.O.S. performed the experiments, F.F.T., A.P., S.E., Y.H.A.L., N.M., B.B.K., and K.S. analyzed the data. F.F.T., B.B.K., and K.S. wrote the manuscript. B.B.K. and K.S. supervised the study.

## Competing interests

The authors declare no competing interests.
