## [Peer Review File · Nature Communications]

REVIEWER COMMENTS

Reviewer #1 (Remarks to the Author):

The DREB2A protein regulates transcription in *A. thaliana* by interactions of its intrinsically disordered activation domain with the negative regulator RCD1 and the Med25 subunit of the Mediator complex. In a very careful and systematic biophysical, NMR and molecular modeling study, F.F. Theisen and colleagues unravel mechanistic details of these interactions. They provide convincing evidence that these interactions (i) are mediated by two distinct short linear motifs in DREB2A, the Med25 ACID-binding sequence (ABS) and the RCD1-interacting motif (RIM), and (ii) depend on the cis/trans conformational equilibrium of proline 245 located in the DREB2A ABS. Thus, the authors feel justified to claim that they have added a new regulatory layer to transcription factor-Mediator interaction by identifying proline isomerization as a new mechanistic principle. Although proline isomerization has been invoked as an event regulating other protein-ligand interactions outside transcription regulation, it is expected that the findings described here will be of considerable interest to the diverse audience of *Nature Communications*.

The extensive data presented in this manuscript are based on considerable prior work of the authors' laboratory on gene regulation in plants and a large new set of meticulous experiments. The manuscript is near-flawlessly written and allows to follow the narrative even in the presence of a large number of acronyms. I have but a small number of very minor comments on issues of presentation.

(1) On page 8, reference to Figure S10 is made before reference to Figure S9. This should be corrected.

(2) Also on page 8: Is "Appendix S1" and "Supplementary Data 1" the same thing? Please clarify.

(3) On page 14, please replace "homologues recombination" with "homologous recombination".

(4) In ref. 39, it should read Spolar, R. S. & Record, M. T.. It's Tom Record, after all!

Reviewer #2 (Remarks to the Author):

Theisen et al. use quantitative binding assays and elegant NMR and biophysical experiments to thoroughly characterize the interaction between *Arabidopsis thaliana* DREB2A transcription factor (TF) and Mediator subunit 25 (Med25). They compare their model for this interaction with previously described human/viral TFs and human MED25. Importantly, their carefully executed and quantitative experiments reveal a bivalent interaction between DREB2A and Med25 similar to those observed for human/viral TF-MED25 pairings. Their results suggest that multivalent TF-mediator interactions are more broadly conserved than previously recognized, with important implications for the evolution of transcriptional regulation. Their research will be of broad interest to many fields (transcriptional regulation, structural biology, intrinsically disordered proteins, etc.) and I recommend its publication in Nature Communications. My comments below are primarily requests for additional elaboration and/or clarification.

Major comments:

1) Page 3 last paragraph and Figure 1:

Please provide further introduction, explanation, and interpretation of the AD tiles experiments in the main text. This likely requires at least its own paragraph. Specifically, from Figure 1 it seems like tiles with the ABS have activity, but only when RIM is also present? Comparing all AD tiles that have RIM, are they more active when they also have ABS than without? Please expand upon interpretation to address these questions. An additional supplementary figure that shows AD tile boundaries (at least for AD1) would help the reader interpret this data for themselves.

2) Page 12, second full paragraph:

In this paragraph the authors make strong statements about how the VP16 H1 and ETV4/5 AD binding sites on MED25 are very different than the DREB2A ABS binding site on Med25. All of these binding sites look similar to me. In general, for all of these cases there are two binding sites on the β -barrel that flank both sides of the α -helices. Please add more description to this paragraph regarding how exactly these binding sites are different.

Modifying Supplementary Figure 15 may help clarify. Which images show CSPs for which interactions? Labels would be helpful here. Are we viewing DREB2A ABS and RIM in all pictures (in red and blue α -helices)? Please add details to the figure legend. Also, further written explanation in the figure legend and in the Discussion of how these models specifically indicate different binding sites would be useful.

In general, given the lack of resolution here, I think the differences need to be really striking to justify plainly stating that they are different. E.g. with high-resolution crystal structures for all of these interactions, it would be easy to spot subtle differences. However all of these MED25-TF studies (not just this one) depend on residue-level resolution in NMR and models of the interactions. There are likely a number of reasons that subtle differences in proposed interactions would be observed (differences in quality of NMR spectra, different modeling programs used, different assumptions used to refine models, etc.). I suggest softening the statement that these binding sites are clearly different, and being more explicit about the limitations of comparing similar dynamic interactions.

As noted above, I think it is quite exciting that *A. thaliana* TF-MED25 interactions suggest a broader conservation of multi- or bivalent interactions between TFs and MED25. I encourage the authors to address this in their discussion.

Minor comments:

1) Page 4, Figure 1 legend

Please add in a short explanation that AD score is reflective of transcriptional activity.

2) Page 4, paragraph 2

It would be useful to have a sequence alignment of Hs and At MED25 as a supplementary figure in this study so that the reader doesn't have to go to a different paper for it.

3) Page 7, paragraph 1

Please explicitly state why AF2 model 3 was discarded. I assume this is due to (clear) disagreement with NMR data. It is worth stating the rationale, whatever it is.

Rationale for AF2 model 2 being more likely is logical. Yet, I think it is also possible that both binding poses may contribute to the overall interaction between DREB2A and Med25. W244A also disrupts binding, though not nearly to the extent as Y242A. A conceptual model of binding modes predicted by AF2 models 1 and 2 both contributing to the overall interaction, with AF2 model 2 being the dominant binding mode, would also be consistent with their biochemical data. If any data rule out this possibility, please explain which. Otherwise I think it is worth including this alternative possibility either here or in the Discussion.

4) Page 11, first sentence of Discussion:

“In this study, we asked how activators and repressors may interact with the same TF through the same binding region.”

This study did not ask how activators and repressors may interact with the same TF through the same binding region. There are no data here for binding to corepressor. Please reword. I suggest first building the overall model from this study, then continuing on to speculation about binding competition with other molecules, etc. Such comments are appropriate, interesting, and useful in the discussion, but should use clear language regarding what was examined in this study and what are comparisons to other studies.

5) General point to incorporate:

Research from Dr. Anna Mapp's group (Henderson et al., 2018 PNAS - link below) suggests that there is allostery between the different binding surfaces on human MED25. Do any of these data allow you to infer whether there is, or isn't, allostery between the ABS and RIM binding sites on Med25? At minimum, it is worth citing this work and stating the possibility of allostery between ABS and RIM binding sites in the Discussion section regarding other MED25-TF interactions, or if your data rule out allostery, explaining why.

<https://www.pnas.org/doi/10.1073/pnas.1806202115>

6) Page 12, third full paragraph:

The authors perform elegant NMR and biophysical experiments and clearly demonstrate that the isomerization of the ABS influences binding to MED25. The difference in trans and cis states in binding to Med25 is fairly subtle (twofold or less changes in binding affinities). Do the authors expect that a larger difference in binding to isomers may exist for the other proteins discussed in the Discussion?

Are there prolyl isomerases in *A. thaliana* for which DREB2A is a known substrate, or at least bind to DREB2A? If so, are there data that indicate how such prolyl isomerases contribute to the stress responses discussed in the introduction? Though the binding difference is subtle, supporting biological data such as this would strengthen the argument for the importance of isomerization. To be clear, I am not asking for any additional experiments, just if any additional context exists in the literature.

Congrats to the authors on a very thorough and interesting study. The experiments are rigorous and elegant, and the visuals in the figures are excellent. I enjoyed diving into this manuscript!

Simon L. Currie

UT Southwestern Medical Center

Reviewer #3 (Remarks to the Author):

This manuscript describes interactions between the ACID domain of the Med25 subunit of the transcriptional co-regulator complex Mediator and different domains of the transcriptional activator protein DREB2A, which is important for the transcriptional response to different types of stresses in *Arabidopsis thaliana*. The results are interesting and of importance for our understanding of the mechanisms that control transcriptional regulation at the molecular level. However, I think the manuscript is written in a way that makes it very difficult for a reader who is not an expert in biophysics. Please find my comments and suggestions below.

Abstract:

-The authors write that the Dreb2a splice variant has retained activity. This is true when it comes to retained activity as a transcriptional activator, but not when it comes to interaction with RCD1. I think the authors should clarify this.

Introduction:

-The authors call Mediator a transcriptional co-activator. I think it is more appropriate to name it as a transcriptional co-regulator, since Mediator is also involved in negative regulation. Even if most studies of Mediator function concern its role as a co-activator, mutations in about half of the Mediator subunits (in yeast) result in up-regulation of transcription.

Results:

-In general, I find that the authors use a terminology that varies and makes it very difficult for the reader. In essence, the study concerns interactions between one domain in Med25 (ACID) and two domains in Dreb2a (ABS and RIM). However, in the text, the authors sometimes refer to ABS as "ABS", and sometimes as "Dreb2a 234-256". It would be much easier to read the text if the authors in the beginning would define that ABS is Dreb2a 234-256, and then use "ABS" in the rest of the paper. At several places the terminology even differs between the figure and its legend. See for example fig. 3A and B where the fragments are labeled "ABS-fragment", "RIM-fragment", and "ABS-RIM-fragment" in figure 3B, but

"DREB2A 234-256", "DREB2A 255-276" and "DREB2A 234-276" in figure 3A. In the legend of the same figure there is a mixture of both versions.

-The results from high-throughput yeast-based assay for screening of ADs is not well described. Figure S3 should be complemented with the AD scores for each bin.

-The text describing figure S4 refers to a core β -barrel with associated helices and the L1-loop. However, there are no indications in the figure showing where these structures are located. It would make it easier to understand the text if the described regions were indicated in figure S3.

-As I am not an expert in NMR, I find it difficult to follow the description of the results presented in fig 2A, 2C, and S5B. In particular, I think that the relation between the MCS SSPs and and ^{13}C SCSs in figure S5B and the relation between the R2/R1s and HetNOEs in figure 2C could be better explained in order for a non-expert to understand the conclusions.

- The text describing figures 3B and 3D includes comparison of CSPs in the L1 loop and the H2 helix. It would be easier to understand the results if the location of these two regions were indicated in figure 3B.

-When describing the results presented in figure S6A, the authors write that comparison of the hetNOEs of free and bound states revealed two regions with significant changes,

one of which mapped to L1 while the other mapped to the C-terminus of H2. I don't understand this conclusion. I find several significant changes in the region between aa 536-~555, while I find that most of the changes in H2 are not more statistically significant than any other region.

-What is the difference between the left and right panels in figure S7? Are they two repeats of the same experiment? It is not described in the figure or in the legend, as far as I can see.

- Figure S10 is described before figure S9.

REVIEWER COMMENTS

Reviewer #1 (Remarks to the Author):

The DREB2A protein regulates transcription in *A. thaliana* by interactions of its intrinsically disordered activation domain with the negative regulator RCD1 and the Med25 subunit of the Mediator complex. In a very careful and systematic biophysical, NMR and molecular modeling study, F.F. Theisen and colleagues unravel mechanistic details of these interactions. They provide convincing evidence that these interactions (i) are mediated by two distinct short linear motifs in DREB2A, the Med25 ACID-binding sequence (ABS) and the RCD1-interacting motif (RIM), and (ii) depend on the cis/trans conformational equilibrium of proline 245 located in the DREB2A ABS. Thus, the authors feel justified to claim that they have added a new regulatory layer to transcription factor-Mediator interaction by identifying proline isomerization as a new mechanistic principle. Although proline isomerization has been invoked as an event regulating other protein-ligand interactions outside transcription regulation, it is expected that the findings described here will be of considerable interest to the diverse audience of Nature Communications.

The extensive data presented in this manuscript are based on considerable prior work of the authors' laboratory on gene regulation in plants and a large new set of meticulous experiments. The manuscript is near-flawlessly written and allows to follow the narrative even in the presence of a large number of acronyms. I have but a small number of very minor comments on issues of presentation.

We appreciate the very positive comments and the suggestions by the reviewer.

(1) On page 8, reference to Figure S10 is made before reference to Figure S9. This should be corrected.

This has been changed.

(2) Also on page 8: Is "Appendix S1" and "Supplementary Data 1" the same thing? Please clarify.

Thank you for pointing out this mistake. Appendix S1 has been changed to Supplementary Method 1 (page 9, second paragraph).

(3) On page 14, please replace "homologues recombination" with "homologous recombination".

This has been changed.

(4) In ref. 39, it should read Spolar, R. S. & Record, M. T.. It's Tom Record, after all!

Thank you for pointing this out. This has of course been changed (now ref. 41).

Reviewer #2 (Remarks to the Author):

Theisen et al. use quantitative binding assays and elegant NMR and biophysical experiments to thoroughly characterize the interaction between *Arabidopsis thaliana* DREB2A transcription factor (TF) and Mediator subunit 25 (Med25). They compare their model for this interaction with previously described human/viral TFs and human MED25. Importantly, their carefully executed and quantitative experiments reveal a bivalent interaction between DREB2A and Med25 similar to those observed for human/viral TF-MED25 pairings. Their results suggest that multivalent TF-mediator interactions are more broadly conserved than previously recognized, with important implications for the evolution of transcriptional regulation. Their research will be of broad interest to many fields (transcriptional regulation, structural biology, intrinsically disordered proteins, etc.) and I recommend its publication in Nature Communications. My comments below are primarily requests for additional elaboration and/or clarification.

We appreciate the extremely positive comments and the many good suggestions given by the reviewer.

Major comments:

1) Page 3 last paragraph and Figure 1:

Please provide further introduction, explanation, and interpretation of the AD tiles experiments in the main text. This likely requires at least its own paragraph. Specifically, from Figure 1 it seems like tiles with the ABS have activity, but only when RIM is also present? Comparing all AD tiles that have RIM, are they more active when they also have ABS than without? Please expand upon interpretation to address these questions. An additional supplementary figure that shows AD tile boundaries (at least for AD1) would help the reader interpret this data for themselves.

Thank you for pointing this out. The activation domain assay is now described in detail in the main text (page 3, paragraph 2). In addition, we have added new panels to Supplementary Fig. 3, which provides further detail on the AD scores of the tiles. Supplementary Fig. 3 now shows that ABS can be included in a tile without significant transcriptional activity. The details of the main text (page 3, paragraph 2) have been updated to help explain the results more comprehensively including the enhancing effect of the ABS on the RIM. We have also updated the corresponding method paragraph.

2) Page 12, second full paragraph:

In this paragraph the authors make strong statements about how the VP16 H1 and ETV4/5 AD binding sites on MED25 are very different than the DREB2A ABS binding site on Med25. All of these binding sites look similar to me. In general, for all of these cases there are two binding sites on the β -barrel that flank both sides of the α -helices. Please add more description to this paragraph regarding how exactly these binding sites are different.

Modifying Supplementary Figure 15 may help clarify. Which images show CSPs for which interactions? Labels would be helpful here. Are we viewing DREB2A ABS and RIM in all pictures (in red and blue α -helices)? Please add details to the figure legend. Also, further written explanation

in the figure legend and in the Discussion of how these models specifically indicate different binding sites would be useful.

We agree that modifying Supplementary Fig. 15 was needed and have done so to improve clarity by separating each interaction into individual panels. Due to the poor NMR spectra often obtained on Med25-ACID (human or *Arabidopsis*) in complex with a bivalent partner, it is difficult to pinpoint binding sites of each individual motif without studying the motifs individually. This could be the reason for not previously identifying the equivalent of the ABS-binding groove in human Med25 interactions. These considerations have been added to the main text (page 12, paragraph 4).

In general, given the lack of resolution here, I think the differences need to be really striking to justify plainly stating that they are different. E.g. with high-resolution crystal structures for all of these interactions, it would be easy to spot subtle differences. However all of these MED25-TF studies (not just this one) depend on residue-level resolution in NMR and models of the interactions. There are likely a number of reasons that subtle differences in proposed interactions would be observed (differences in quality of NMR spectra, different modeling programs used, different assumptions used to refine models, etc.). I suggest softening the statement that these binding sites are clearly different, and being more explicit about the limitations of comparing similar dynamic interactions.

Thank you for highlighting this issue, which we did not properly address. The most striking observation is that, to our knowledge, no previous studies have identified the groove located between loop 1 and helix 2 as a binding site for the N-terminal motif of the bivalent interaction region (of VP16, p53, ERM etc.). We have made this argument clearer in the main text by adding a sentence to soften the statement slightly, highlighting the drawbacks of NMR experiments using bivalent interaction fragments (page 12, paragraph 4).

As noted above, I think it is quite exciting that *A. thaliana* TF-MED25 interactions suggest a broader conservation of multi- or bivalent interactions between TFs and MED25. I encourage the authors to address this in their discussion.

Thank you for this suggestion, we have now mention this in the discussion (page 12, paragraph 4).

Minor comments:

1) Page 4, Figure 1 legend

Please add in a short explanation that AD score is reflective of transcriptional activity.

We have now in more detail explained the activation assay in the main text (page 3, paragraph 3). Furthermore, a sentence has been added to the legend of Fig. 1.

2) Page 4, paragraph 2

It would be useful to have a sequence alignment of Hs and At MED25 as a supplementary figure in this study so that the reader doesn't have to go to a different paper for it.

This is a good idea. A sequence alignment of human and *Arabidopsis* Med25-ACID has been added to Supplementary Fig. 4, which is referred to in the main text (page 4, paragraph 1).

3) Page 7, paragraph 1

Please explicitly state why AF2 model 3 was discarded. I assume this is due to (clear) disagreement with NMR data. It is worth stating the rationale, whatever it is.

We now elaborate on the explanation of why model 3 was discarded (page 7, paragraph 1).

Rationale for AF2 model 2 being more likely is logical. Yet, I think it is also possible that both binding poses may contribute to the overall interaction between DREB2A and Med25. W244A also disrupts binding, though not nearly to the extent as Y242A. A conceptual model of binding modes predicted by AF2 models 1 and 2 both contributing to the overall interaction, with AF2 model 2 being the dominant binding mode, would also be consistent with their biochemical data. If any data rule out this possibility, please explain which. Otherwise I think it is worth including this alternative possibility either here or in the Discussion.

The possibility of alternative binding modes is intriguing, and we agree that these cannot be excluded. For the ABS, we don't see any indications that it may bind the same site in multiple poses with any significant population. This would have resulted in poor NMR peak intensities during the titrations. In addition, the single exponential decay observed for both association and dissociation of the ABS fragment suggests similar kinetics of all potential bound states, and both ITC and NMR suggest 1:1 binding, indicating very poor affinity for any non-overlapping alternative sites. Another strong indication for a single primary binding mode is that the Med25-ACID chemical shifts in complex with ABS and ABS-RIM are very similar around the ABS binding groove (Fig. 3A). If different binding modes were significantly populated, the inclusion of the RIM would likely affect these differently, which is inconsistent with similar chemical shifts of ABS-groove residues.

We now mentioned this more directly in the main text (page 6, paragraph 2). Furthermore, we have softened the conclusion, so that we do not rule out the potential existence of minor states at certain conditions (page 8, paragraph 1). Furthermore, in Discussion we now open for conditional alternative binding modes/configurations (page 12, paragraph 4).

4) Page 11, first sentence of Discussion:

"In this study, we asked how activators and repressors may interact with the same TF through the same binding region."

This study did not ask how activators and repressors may interact with the same TF through the same binding region. There are no data here for binding to corepressor. Please reword. I suggest first building the overall model from this study, then continuing on to speculation about binding competition with other molecules, etc. Such comments are appropriate, interesting, and useful in the discussion, but should use clear language regarding what was examined in this study and what are comparisons to other studies.

We agree with this comment. The first sentence of the discussion has been modified to clarify that the work presented here focus on the interaction between Med25 and DREB2A. We, however, refer to the interacting region of DREB2A as a hot spot to point out its interaction with another coregulator (page 12, paragraph 2).

5) General point to incorporate:

Research from Dr. Anna Mapp's group (Henderson et al., 2018 PNAS - link below) suggests that there is allostery between the different binding surfaces on human MED25. Do any of these data allow you to infer whether there is, or isn't, allostery between the ABS and RIM binding sites on Med25? At minimum, it is worth citing this work and stating the possibility of allostery between ABS and RIM binding sites in the Discussion section regarding other MED25-TF interactions, or if your data rule out allostery, explaining why.

This is an interesting aspect that we only briefly mention. Based on the RIM affinity of a K_d around 5 mM and the linker length, we would expect a RIM saturation to be around 65%. However, the CEST analysis found that for a sample containing Med25ACID:DREB2A targeting a 5:100 molar ratio, 4.6% of the ABS was bound and 4.7% of the RIM was bound. This could suggest an allosteric effect resulting from the RIM in the ABS bound state or linker binding. This was mentioned on page 9 paragraph 1 but has now been rephrased with the inclusion of a reference to the paper by the Mapp group.

We find that that a more elaborate discussion of this would be too speculative. We are currently establishing experimental systems for studies of additional binding partners with different binding surfaces, which will follow up on this issue.

6) Page 12, third full paragraph:

The authors perform elegant NMR and biophysical experiments and clearly demonstrate that that the isomerization of the ABS influences binding to MED25. The difference in trans and cis states in binding to Med25 is fairly subtle (twofold or less changes in binding affinities). Do the authors expect that a larger difference in binding to isomers may exist for the other proteins discussed in the Discussion?

The effect on affinity is fairly small and although the effect is larger for the kinetic parameters (k_{off}) we can only speculate on the biological relevance of proline isomerization. However, and as suggested in the last paragraph of Discussion and in our model figure 7, the isomerization appears to confer structural heterogeneity which may affect ternary complex formation. At the moment, we do not have sufficient data for other binding partners to infer effects of proline isomerization on these interactions.

Are there prolyl isomerases in *A. thaliana* for which DREB2A is a known substrate, or at least bind to DREB2A? If so, are there data that indicate how such prolyl isomerases contribute to the stress responses discussed in the introduction? Though the binding difference is subtle, supporting biological data such as this would strengthen the argument for the importance of isomerization. To be clear, I am not asking for any additional experiments, just if any additional context exists in

the literature.

This is a highly relevant question, which we are also considering. Recent studies have shown expansion of cyclophilins in plants compared to animals and prokaryotes, and several of these possess peptidyl-prolyl *cis-trans* isomerases activity. Although several of the cyclophilins are also implicated in stress responses, as is also the case for DREB2A, DREB2A has so far not been identified as a substrate for any of these. In the revised manuscript, we briefly mention this in Discussion (page 13, paragraph 1), also as a future research question.

Congrats to the authors on a very thorough and interesting study. The experiments are rigorous and elegant, and the visuals in the figures are excellent. I enjoyed diving into this manuscript!

Thank you again for the favorable comments. This is highly appreciated.

Simon L. Currie
UT Southwestern Medical Center

Reviewer #3 (Remarks to the Author):

This manuscript describes interactions between the ACID domain of the Med25 subunit of the transcriptional co-regulator complex Mediator and different domains of the transcriptional activator protein DREB2A, which is important for the transcriptional response to different types of stresses in *Arabidopsis thaliana*. The results are interesting and of importance for our understanding of the mechanisms that control transcriptional regulation at the molecular level. However, I think the manuscript is written in a way that makes it very difficult for a reader who is not an expert in biophysics. Please find my comments and suggestions below.

We thank the reviewer for the appreciation of the importance of our work. We also highly appreciate the reviewers perspective and considerations for non-biophysics readers and welcome the many good suggestions for making the manuscript more accessible to a broader audience.

Abstract:

-The authors write that the Dreb2a splice variant has retained activity. This is true when it comes to retained activity as a transcriptional activator, but not when it comes to interaction with RCD1. I think the authors should clarify this.

The abstract has been updated to emphasize that activity refers to transcriptional activity.

Introduction:

-The authors call Mediator a transcriptional co-activator. I think it is more appropriate to name it as a transcriptional co-regulator, since Mediator is also involved in negative regulation. Even if most studies of Mediator function concern its role as a co-activator, mutations in about half of the Mediator subunits (in yeast) result in up-regulation of transcription.

We agree with the reviewer and have changed the classification of Med25 to co-regulator.

Results:

-In general, I find that the authors use a terminology that varies and makes it very difficult for the reader. In essence, the study concern interactions between one domain in Med25 (ACID) and two domains in Dreb2a (ABS and RIM). However, in the text, the authors sometimes refers to ABS as "ABS", and sometimes as "Dreb2a 234-256". It would be much easier to read the text if the authors in the beginning would define that ABS is Dreb2a 234-256, and then use "ABS" in the rest of the paper. At several places the terminology even differs between the figure and its legend. See for example fig. 3A and B where the fragments are labeled "ABS-fragment", "RIM-fragment", and "ABS-RIM-fragment" in figure 3B, but "DREB2A 234-256", "DREB2A 255-276" and "DREB2A 234-276" in figure 3A. In the legend of the same figure there is a mixture of both versions.

We thank the reviewer for pointing this out and have now adopted the motif naming scheme throughout the main text, legends and supplementary information. ABS, RIM and ABS-RIM are defined in Table 1 and on page 5 paragraph 3 and 4, and page 6 paragraph 3. Thus, we have changed the main text (e.g., page 9, paragraph 1), and we have updated the legend for figure 3, implementing the ABS, RIM and ABS-RIM naming. Furthermore, we have reevaluated the use of the word 'fragment' when associated with the named fragments (e.g. page 9, paragraph 4).

-The results from high-throughput yeast-based assay for screening of ADs is not well described. Figure S3 should be complemented with the AD scores for each bin.

We agree with this (see also comment for reviewer #2). We have now expanded Supplementary Fig. 3 to show the assay results in more detail, including the binning fluorescence ratio key, bin scores/weights, tile abundancy and individual tile boundaries for DREB2A. In addition, we have introduced a new paragraph in the main text explaining the assay and the results in more detail (page 3, paragraph 2), and expanded the associated Method paragraph accordingly.

-The text describing figure S4 refers to a core β -barrel with associated helices and the L1-loop. However, there are no indications in the figure showing where these structures are located. It would make it easier to understand the text if the described regions were indicated in figure S3.

Labels have now been added to Supplementary Fig. 4.

-As I am not an expert in NMR, I find it difficult to follow the description of the results presented in fig 2A, 2C, and S5B. In particular, I think that the relation between the MCS SSPs and and ^{13}C SCSs in figure S5B and the relation between the R2/R1s and HetNOEs in figure 2C could be better explained in order for a non-expert to understand the conclusions.

Thank you for pointing this out. We agree that it was not explained well. We have rephrased the main text (page 4, paragraph 1) to emphasize that one method (SCSs) provides an estimate of the secondary structure based on single nuclei, while the other (MICS) incorporates data from multiple

nuclei. In addition, we have added a brief description of the SCS and MICS analyses in the legend of Supplementary Fig. 5.

Regarding the R_2/R_1 and hetNOE analysis, both methods provide information on the dynamics of the backbone amide, where the R_2/R_1 in addition are sensitive to chemical exchange. We have added additional explanatory text to the main text including a parenthesis informing how to interpret the hetNOEs (page 5, paragraph 1). For the Figure 2 legend, we have added a sentence describing the relation between fast timescale dynamics and R_2/R_1 , hetNOEs and MD RMSF.

- The text describing figures 3B and 3D includes comparison of CSPs in the L1 loop and the H2 helix. It would be easier to understand the results if the location of these two regions were indicated in figure 3B.

We have added a secondary structure cartoon above panel B showing the location of the secondary structures. In addition, we have increased the visibility of markings highlighting residues of interest, such as Gly570 and Phe623.

-When describing the results presented in figure S6A, the authors write that comparison of the hetNOEs of free and bound states revealed two regions with significant changes, one of which mapped to L1 while the other mapped to the C-terminus of H2. I don't understand this conclusion. I find several significant changes in the region between aa 536~555, while I find that most of the changes in H2 are not more statistically significant than any other region.

Thank you for pointing this out. Residues preceding 555 are very dynamic, even in the bound state. We have now in the main text clarified that we are focusing only on residues within the folded ACID domain (page 5, paragraph 1).

-What is the difference between the left and right panels in figure S7? Are they two repeats of the same experiment? It is not described in the figure or in the legend, as far as I can see.

Yes, it simply shows the repeat of the experiment yielding similar parameters. This is now described in the legend of Supplementary Fig. 7.

- Figure S10 is described before figure S9.

The order has been reversed, thank you.

REVIEWERS' COMMENTS

Reviewer #2 (Remarks to the Author):

The authors have addressed all of my minor concerns. I enthusiastically recommend publication of the revised manuscript in Nature Communications.